# Epidemiological investigation and physician awareness regarding the diagnosis and management of Q fever in South Korea, 2011 to 2017

Yong Chan Kim[1], Hye Won Jeong[2], Dong-Min Kim[3], Kyungmin Huh[4], Sang-Ho Choi[5], Hee Young Lee[6], Yunjung Jung[7], Yeol Jung Seong[8], Eun Jin Kim[8], Young Hwa Choi[8], Jung Yeon Heo[8]*

1 Division of Infectious Diseases, Department of Internal Medicine, Yonsei University College of Medicine, Seoul, Republic of Korea, 2 Division of Infectious Diseases, Department of Internal Medicine, Chungbuk National University College of Medicine, Cheongju, Republic of Korea, 3 Division of Infectious Diseases, Department of Medicine, Chosun University College of Medicine, Kwangju, Republic of Korea, 4 Division of Infectious Diseases, Department of Medicine, Samsung Medical Center, Sungkyunkwan University School of Medicine, Seoul, Republic of Korea, 5 Department of Infectious Diseases, Asan Medical Center, University of Ulsan College of Medicine, Seoul, Republic of Korea, 6 Center for Preventive Medicine and Public health, Seoul National University Bundang Hospital, Seongnam, Republic of Korea, 7 Department of Pulmonology and Critical Care Medicine, Ajou University School of Medicine, Suwon, Republic of Korea, 8 Department of Infectious Diseases, Ajou University School of Medicine, Suwon, Republic of Korea

* jyeon78@naver.com

**Data Availability Statement:** All relevant data are within the manuscript and its Supporting Information files.

## Abstract

### Background

In South Korea, the number of Q fever cases has rapidly increased since 2015. Therefore, this study aimed to characterize the epidemiological and clinical features of Q fever in South Korea between 2011 and 2017.

### Methods/Principal findings

We analyzed the epidemiological investigations and reviewed the medical records from all hospitals that had reported at least one case of Q fever from 2011 to 2017. We also conducted an online survey to investigate physicians' awareness regarding how to appropriately diagnose and manage Q fever. The nationwide incidence rate of Q fever was annually 0.07 cases per 100,000 persons. However, there has been a sharp increase in its incidence, reaching up to 0.19 cases per 100,000 persons in 2017. Q fever sporadically occurred across the country, with the highest incidences in Chungbuk (0.53 cases per 100,000 persons per year) and Chungnam (0.27 cases per 100,000 persons per year) areas. Patients with acute Q fever primarily presented with mild illnesses such as hepatitis (64.5%) and isolated febrile illness (24.0%), whereas those with chronic Q fever were likely to undergo surgery (41.2%) and had a high mortality rate (23.5%). Follow-up for 6 months after acute Q fever was performed by 24.0% of the physician respondents, and only 22.3% of them reported that clinical and serological evaluations were required after acute Q fever diagnosis.

**Funding:** This work was supported by grants funded by the Korea Centers for Disease Prevention and Control (Grant No. 2018E230100; J.Y.H.). The funders had no role in study design, data collection and analysis, decision to publish, or preparation of the manuscript.

**Competing interests:** The authors have declared that no competing interests exist.

## Conclusions

Q fever is becoming an endemic disease in the midwestern area of South Korea. Given the clinical severity and mortality of chronic Q fever, physicians should be made aware of appropriate diagnosis and management strategies for Q fever.

## Author summary

Human Q fever, zoonosis caused by *Coxiella burnetii*, presents with diverse clinical manifestations, from self-limited febrile illnesses to endocarditis. It is usually diagnosed using serological tests. Because of the diagnostic challenge and nonspecific symptoms, it has been considered an underrecognized infectious disease, particularly in non-endemic or non-epidemic areas. In South Korea, Q fever was designated as a notifiable disease in 2006, and the number of Q fever cases has increased sharply since 2015. This study shows that Q fever is becoming an endemic disease in the midwestern area of South Korea. In this study, patients with acute Q fever primarily presented with mild illnesses such as hepatitis (64.5%) and isolated febrile illness (24.0%). However, patients with chronic Q fever presented with vascular infection (52.9%) and endocarditis (35.3%); such patients were likely to undergo surgery (41.2%) and had a high mortality rate (23.5%). We also surveyed physicians' awareness regarding how to appropriately diagnose and manage Q fever. This survey showed that more than half of the physicians did not follow-up suspected patients with acute Q fever who had negative serological results at the early stage of illness using a serological test and that follow-ups for identifying chronic Q fever after primary *C. burnetii* infection were only performed by 24.0% of physicians.

## Introduction

Human Q fever is a worldwide zoonosis caused by *Coxiella burnetii*. It presents with diverse clinical manifestations, from self-limited febrile illnesses to endocarditis. Although transmission to humans can occur through direct contact with infected animals, the primary route of transmission is inhalation of the organism through aerosol particles from a contaminated environment [1]. Therefore, Q fever can occur in community-living individuals who are not exposed to well-known risk factors, such as contact with animals or occupational exposures, and has resulted in large-scale outbreaks [2].

Q fever is usually diagnosed using serological tests in patients with relevant symptoms of *C. burnetii* infection. Indirect immunofluorescence antibody (IFA) tests, in which the paired serum samples are obtained 3–6 weeks apart, are used to confirm the diagnosis [3]. Thus, Q fever has long been considered an underdiagnosed and underreported disease [4,5]. Although acute Q fever is usually benign with a low mortality rate, a large-scale outbreak of Q fever can lead to public health concerns such as blood transfusion-related transmission or obstetrical complications in pregnant women. Furthermore, chronic Q fever has been observed as a persistent localized infection in <5% of patients with *C. burnetii* infections. Chronic Q fever is often difficult to treat, and it can be fatal in 5%–50% of patients, if left untreated [3,6].

In South Korea, Q fever was designated as a mandatory notifiable infectious disease in 2006. Between 2006 and 2011, the number of annual cases of Q fever reported to the Korean Centers for Disease Control and Prevention (KCDC) was approximately 10 [7]. Since 2015, the number of reported cases has increased rapidly. However, there are limited data regarding

the recent epidemiological and clinical characteristics of patients infected with *C. burnetii* in South Korea. Therefore, this study aimed to characterize the epidemiological and clinical features of Q fever in South Korea between 2011 and 2017. In addition, we surveyed physicians' awareness regarding the disease to determine whether patients suspected of having Q fever were being accurately diagnosed and whether confirmed patients were followed up to assess the status of the disease after primary infection.

## Methods

### Ethics statement

This study was approved by the Institutional Review Board of Ajou University Hospital. This was a retrospective study using anonymized, aggregated data from the KCDC and the included hospitals. Since this study did not have a harmful influence on the patients or the institutions, the Institutional Review Board waived the requirement for obtaining written consent from the patients.

### Study design and data collection

A retrospective descriptive study was conducted based on the Q fever epidemiological investigation reports, which were submitted to the KCDC from 2011 to 2017. These reports included data on age, sex, residential area, date of symptom onset, date of diagnosis, and behavioral risk factors such as contact with animals or occupational exposure to *C. burnetii*. To collect clinical information, we reviewed medical records directly from all 26 hospitals that had reported at least one case of Q fever during the study period. Clinical data on the following were collected from the 26 hospitals: age, sex, date of symptom onset, date of first hospital visit, underlying comorbidities, pregnancy status, initial clinical manifestations, laboratory test results, prescribed antibiotics, and treatment outcomes.

### Diagnosis of Q fever

Based on the laboratory criteria, acute Q fever was defined as confirmed or probable in patients with symptoms consistent with acute febrile illness such as flu-like symptoms, hepatitis, or pneumonia. Laboratory-confirmed cases of acute Q fever were defined as follows: 1) a $\geq$4-fold increase in immunoglobulin G (IgG) titer to phase II antigen between paired serum samples or 2) positive polymerase chain reaction (PCR) results for *C. burnetii* nucleic acid in the clinical specimens. If the IgG titer to phase II antigen in the acute phase was $\geq$1:256 and the follow-up serum sample from the convalescent phase was not collected, the case was considered a laboratory probable case. Chronic Q fever was diagnosed based on serological test, PCR, or culture results in patients with the following clinical presentations: culture-negative endocarditis, vascular infection, osteomyelitis, osteoarthritis, or chronic organ infection without specific etiology. The serological criteria for the diagnosis of chronic Q fever were a single IgG titer to phase I antigen $\geq$1:800 and a higher antibody titer to phase I antigen than to phase II antigen.

Serological tests for diagnosing *C. burnetii* infection were performed at the KCDC using IFA assay kits (Focus diagnostics, Cypress, CA, USA). Molecular diagnosis was performed using real-time PCR with the *C. burnetii* IS*1111* gene in individual laboratories.

### Survey to evaluate physicians' awareness regarding Q fever

We administered an online survey to physicians, including infectious disease specialists, who were registered with the Korean Society of Infectious Disease from December 17 to December

31, 2018. The survey consisted of questions to investigate physicians' awareness regarding how to accurately diagnosis and manage Q fever. The questionnaire contained 12 questions regarding physicians' clinical careers, place of work, experience in treating patients with Q fever, and knowledge regarding its diagnosis and management. The detailed questions and their answers are presented in S1 Appendix.

### Data analysis

The epidemiological data provided by the KCDC were used to analyze the spatial and temporal distribution of Q fever in South Korea. To describe the spatial distribution of disease, we used the incidence rates expressed as the total number of cases per 100,000 persons at the national and regional levels. The average number of inhabitants in the regions was obtained from the data provided by the Korean Statistical Information Service database [8].

Clinical data are expressed as number with percentage for categorical variables and median with interquartile range (IQR) for continuous variables. To determine the differences between baseline characteristics of patients with acute and chronic Q fever, Chi-square test was used for the categorical variables and Mann–Whitney test was used for the continuous variables. Cases in which the clinical data were unavailable from the hospitals were only included in the analysis of the spatial and temporal distribution using only the data from the KCDC.

## Results

### Nationwide incidence of Q fever in South Korea

A total of 241 Q fever cases were reported to the KCDC from 2011 to 2017. The cases occurred sporadically across the country. The median age of patients with Q fever was 54.5 years (IQR 47.0–62.0 years), with 70 (35.0%), 46 (23.0%), and 37 (18.5%) patients in the 50s, 60s, and 40s age groups, respectively (S1 Fig). The nationwide incidence rate from 2011 to 2017 was 0.07 cases per 100,000 persons per year. However, there was a sharp increase from 0.02 cases per 100,000 persons per year in 2011 to 0.05–0.19 cases per 100,000 persons per year after 2015. The highest incidence was observed in Chungbuk (0.53 cases per 100,000 persons per year), followed by Chungnam (0.27 cases per 100,000 persons per year) (Fig 1). Although Q fever case were most commonly reported in summer (32.0%) and autumn (30.3%), cases were reported throughout the year (S1 Table).

### Epidemiological characteristics of patients with Q fever

Of the 241 patients reported to the KCDC, 200 (83%) patients whose data were available in the hospital databases were included in the analysis of the epidemiological characteristics of Q fever. Table 1 shows a comparison of the baseline characteristics of patients with acute and chronic Q fever in South Korea. The median age of patients with acute Q fever was 54.0 years (IQR 46.0–62.0 years), and there were 165 (90.2%) male patients. Among the 183 patients with acute Q fever, 45 (24.6%) patients had occupational risks. In addition, 43 (23.5%) patients had a history of animal contact (of any kind). Although the most common kinds of animal contact were domestic ruminants such as goats (21 cases, 11.5%) and cattle (13 cases, 17.1%), some patients had a history of contact with a wide range of companion animals and livestock, such as dogs (3 cases, 1.6%) and pigs (1 case, 0.5%). In total, 85 (46.4%) patients with acute Q fever lived in rural areas, and they were likely to live near livestock farms. Epidemiological factors associated with Q fever, such as occupation, history of animal contact, and living area, were identified in only 88 (48.1%) cases of acute Q fever. There was no statistically significant difference in the epidemiological characteristics between patients with acute and chronic Q fever

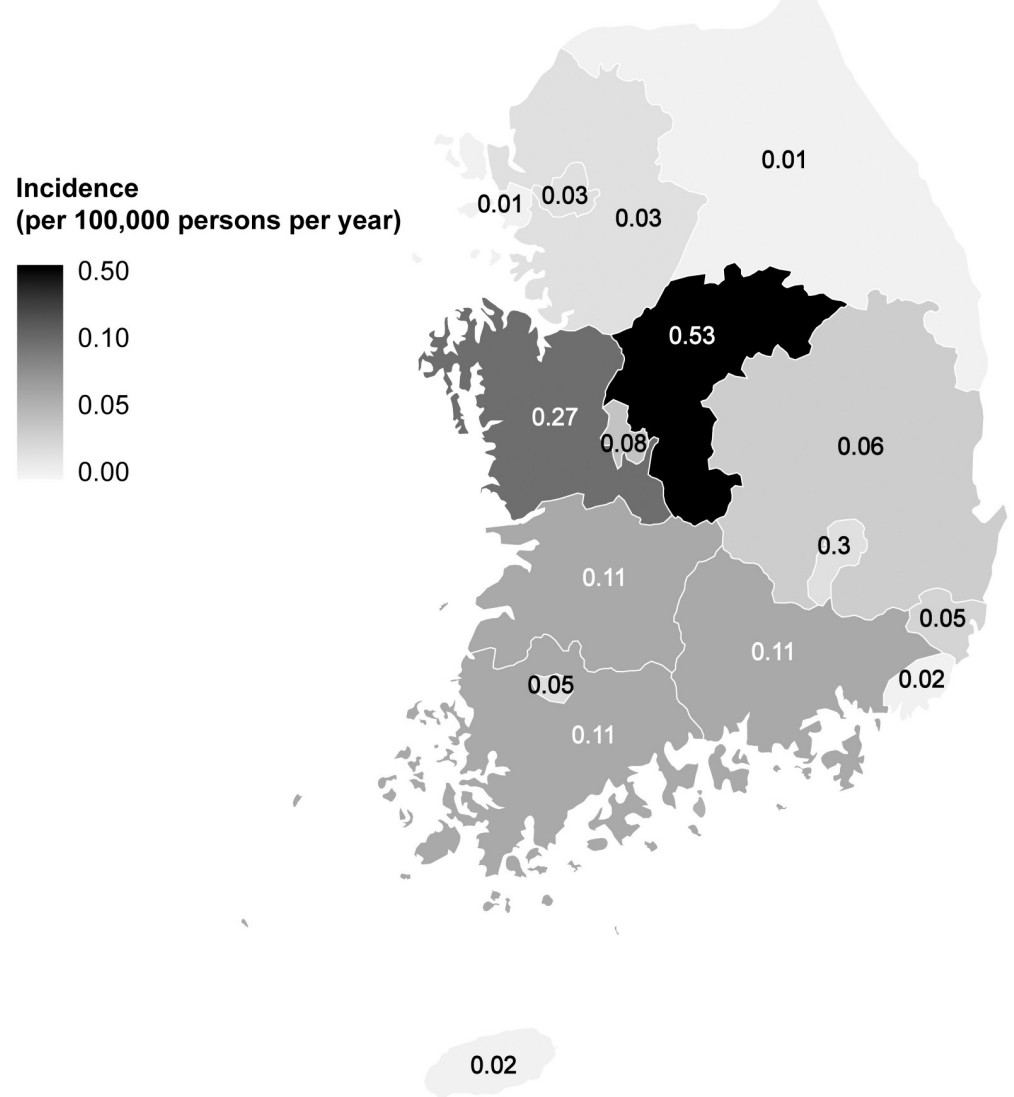

**Fig 1. Geographic distribution of patients with Q fever in South Korea from 2011 to 2017.** Source: https://www.ngii.go.kr/eng/main.do.

($P$ > 0.05). However, patients with chronic Q fever were more likely to have congestive heart failure ($P$ = 0.001), heart valve insufficiency ($P$ < 0.001), prosthetic heart valves ($P$ = 0.037), and arterial aneurysms ($P$ < 0.001) than patients with acute Q fever.

### Clinical characteristics of patients with acute Q fever

Patients with acute Q fever commonly presented with nonspecific symptoms, including fever (n = 175, 95.6%), myalgia (n = 107, 58.5%), and chills (n = 101, 55.2%) (Table 2). Initial laboratory findings showed modestly elevated levels of C-reactive protein (7.9 mg/dL [IQR 2.6–12.3 mg/dL]) and alanine aminotransferase (74 IU/L [IQR 37–128 IU/L]), with barely normal ranges of white blood cells (7,920 /mm$^3$ [IQR 5,287–11,202 /mm$^3$]). The clinical manifestations were primarily hepatitis (118 cases, 64.5%), isolated febrile illness (44 cases, 24.0%), and

**Table 1. Baseline characteristics of patients with acute and chronic Q fever in South Korea from 2011 to 2017.**

| | Acute Q fever (n = 183) | Chronic Q fever (n = 17) | P value |
|---|---|---|---|
| | No. (%) | | |
| Male | 165 (90.2) | 15 (88.2) | 0.681 |
| Age, years (median, IQR) | 54.0 (46.0–62.0) | 58.0 (51.5–68.0) | 0.643 |
| Occupation | 45 (24.6) | 3 (17.6) | 0.767 |
| Agricultural farmer | 25 (13.7) | 3 (17.6) | |
| Livestock raiser | 17 (9.3) | 0 (0) | |
| Veterinarian | 2 (1.1) | 0 (0) | |
| Slaughterhouse worker | 1 (0.5) | 0 (0) | |
| Others | 138 (75.4) | 14 (82.4) | |
| Animal contact | 43 (23.5) | 3 (17.6) | 0.767 |
| Goat | 21 (11.5) | 0 (0) | |
| Cattle | 13 (7.1) | 3 (17.6) | |
| Dog | 3 (1.6) | 0 (0) | |
| Rabbit | 2 (1.1) | 0 (0) | |
| Deer | 1 (0.5) | 0 (0) | |
| Hoarse | 1 (0.5) | 0 (0) | |
| Cat | 1 (0.5) | 0 (0) | |
| Pig | 1 (0.5) | 0 (0) | |
| Living area: rural | 85 (46.4) | 6 (35.3) | 0.377 |
| Underlying diseases | 69 (37.7) | 12 (70.6) | 0.008 |
| Hypertension | 44 (24.0) | 3 (17.6) | 0.767 |
| Diabetes mellitus | 38 (20.8) | 2 (11.8) | 0.533 |
| Congestive heart failure | 6 (3.3) | 5 (29.4) | 0.001 |
| Heart valve insufficiency | 0 (0) | 5 (29.4) | <0.001 |
| Heart valve prosthesis | 2 (1.1) | 2 (11.8) | 0.037 |
| Arterial aneurysm | 1 (0.5) | 6 (35.3) | <0.001 |
| Chronic pulmonary disorder | 1 (0.5) | 0 (0) | 1 |
| Malignancy | 4 (2.2) | 0 (0) | 1 |
| Hospitalization | 152 (83.1) | 13 (76.5) | 0.506 |
| Illness onset to hospital visit, (median, IQR) | 7 (4–14) | 10 (5–16) | 0.425 |

IQR: interquartile range.

pneumonia with or without hepatitis (17 cases, 9.3%). Among the patients with acute Q fever, 116 (63.4%) patients underwent serological tests using paired sera collected at least twice during the acute and convalescent phase. Therefore, the remaining 67 (36.6%) cases in which only a single serological test was performed were classified as probable acute Q fever cases. Fifty (27.3%) patients with acute Q fever had seroconversion, from seronegative in the acute stage to seropositive in the convalescent stage. After primary infection, serological follow-ups and echocardiography were performed in 75 (41.0%) and 113 (61.7%) cases, respectively. Positive PCR results for *C. burnetii* were identified in seven cases that showed negative serological results in the acute stage. Clinical and serological follow-ups at 6 months after primary infection were only performed in 44 (24.0%) patients with acute Q fever. Appropriate antibiotic treatment, including doxycycline, trimethoprim-sulfamethoxazole, fluoroquinolone, or macrolide antibiotics, was administered in 158 (86.3%) cases, and the median duration of antibiotic treatment was 14.0 days (IQR 9.8–17.0 days). None of the patients with acute Q fever progressed to persistent localized infection or died.

**Table 2. Clinical characteristics of patients with acute Q fever in South Korea from 2011 to 2017.**

| Variables | Acute Q fever (n = 183) |
|---|---|
| | No. (%) |
| Clinical symptoms | |
| Fever | 175 (95.6) |
| Chill | 101 (55.2) |
| Myalgia | 107 (58.5) |
| Headache | 88 (48.1) |
| Fatigue/weakness | 55 (30.1) |
| Cough | 36 (19.7) |
| Laboratory findings | |
| WBC (/mm$^3$), median (IQR) | 7,920 (5,287–11,202) |
| Platelet (×10$^3$/mm$^3$), median (IQR) | 199 (127–371) |
| CRP (mg/dL), median (IQR) | 7.9 (2.6–12.3) |
| ALT (IU/L), median (IQR) | 74 (37–128) |
| LDH (IU/L), median (IQR) | 399 (272–587) |
| Diagnostic test | |
| Initial seronegative result | 50 (27.3) |
| Positive PCR | 7/29 (24.1) |
| Clinical manifestations | |
| Hepatitis | 118 (64.5) |
| Isolated febrile illness | 44 (24.0) |
| Pneumonia with/without hepatitis | 17 (9.3) |
| Meningoencephalitis | 3 (1.6) |
| Pericarditis | 1 (0.5) |
| Case classification* | |
| Confirmed | 116 (63.4) |
| Probable | 67 (36.6) |
| Follow-up after primary infection | |
| Serological test ≥3 | 75 (41.0) |
| Serological test ≥4 | 43 (23.5) |
| Echocardiography | 113 (61.7) |
| 6-month follow-up | 44 (24.0) |
| Prognosis | |
| Chronic Q fever after primary infection | 0 (0) |
| Treatment duration (day), median (IQR) | 14.0 (9.8–17.0) |

IQR, interquartile range; WBC, white blood cell; CRP, C-reactive protein; ALT, alanine aminotransferase; LDH, lactate dehydrogenase.

*Confirmed or probable cases were classified using the laboratory criteria with clinical evidence of *C. burnetii* infection.

## Clinical characteristics of patients with chronic Q fever

Of the 200 patients with Q fever enrolled in this study, 17 patients were classified as having chronic Q fever. Vascular infection (8 cases, 47.1%) was the most frequent manifestation of chronic Q fever (Table 3). Of these 8 cases, 3 cases had aortic prosthesis-related infections and the other 5 cases had an infected aortic aneurysm. Six (37.5%) patients had endocarditis. Surgical treatment, such as valve replacement or vascular prosthesis removal, was performed in 7

**Table 3. Clinical characteristics of patients with chronic Q fever in South Korea from 2011 to 2017.**

| Variables | Chronic Q fever (n = 17) |
|---|---|
| | No. (%) |
| Clinical symptoms | |
| Fever | 9 (52.9) |
| Chill | 5 (29.4) |
| Myalgia | 4 (23.5) |
| Headache | 4 (23.5) |
| Fatigue/weakness | 5 (29.4) |
| Laboratory findings | |
| WBC (/mm$^3$), median (IQR) | 8,380 (5,703–12,570) |
| Platelet ($\times 10^3$/mm$^3$), median (IQR) | 235 (129–331) |
| CRP (mg/dL), median (IQR) | 7.4 (3.5–10.0) |
| ALT (IU/L), median (IQR) | 56 (25–160) |
| LDH (IU/L), median (IQR) | 402 (259–489) |
| Clinical manifestations | |
| Vascular infection | 8 (47.1) |
| Endocarditis | 6 (35.3) |
| Bone and joint infection | 3 (17.6) |
| Prognosis | |
| Surgical treatment | 7 (41.2) |
| Treatment duration (month), median (IQR) | 24.0 (13.5–40.0) |

IQR, interquartile range; WBC, white blood cell; CRP, C-reactive protein; ALT, alanine aminotransferase; LDH, lactate dehydrogenase.

(41.2%) cases. The median duration of antibiotic treatment was 24.0 months (IQR 13.5–40.0 months) for patients with chronic Q fever. The mortality rate was 23.5%.

## Survey to evaluate the physician's awareness regarding Q fever

We surveyed 1,226 physicians online, including 243 infectious disease specialists, and the response rate was 8.4%. The questionnaire results are presented in Table 4. Most physicians (80.6%) worked in university-affiliated hospitals, and 86.4% of respondents had experience performing Q fever tests. However, 38.7% of respondents had no experience in diagnosing Q fever, and 35.0% of respondents had only experience of one or two cases of Q fever. When patients were suspected of having acute Q fever, 55.3% of respondents rarely conducted a second serological test in the convalescent stage if the sera from the acute stage was negative. With regard follow-up clinical assessments and serological tests repeated over 6 months to identify cases of chronic Q fever after acute Q fever, 40.8% of respondents stated that clinical assessments and serological tests were not repeated after primary infection, while 22.3% of respondents repeated clinical assessments and serological tests. The other questions and their results are presented in S2 Table.

## Discussion

This study shows that Q fever is becoming an endemic disease in the midwestern area of South Korea. Although the incidence rate of Q fever was as low as 0.07 cases per 100,000 persons per year across the county, the annual incidence rate has risen sharply, reaching up to 0.19 cases per 100,000 persons in 2017. Additionally, some areas such as Chungbuk (0.53 cases per

**Table 4. Physician's awareness regarding the diagnosis and management of Q fever.**

| Questions and answers | Response rate (%) |
|---|---|
| Q1) Have you ever requested a Q fever antibody or PCR test? How many patients did you test for Q fever? | |
| Not at all | 13.6 |
| <10 cases | 43.7 |
| 10–30 cases | 29.1 |
| >30 cases | 13.6 |
| Q2) For which patients did you request a Q fever test? (select all applicable items) | |
| Never requested a Q fever test | 11.7 |
| A febrile patient with an occupational risk, such as livestock raisers | 40.8 |
| A febrile patient with a history of animal contact, such as contact with cattle, sheep, or goats | 45.6 |
| A febrile patient with a history of visiting a barn or farm | 31.1 |
| A patient who has fever of unknown origin >3 weeks without any epidemiological risk | 60.2 |
| A patient who has unexplained fever within 2 weeks | 46.6 |
| A patient with fever and modestly elevated transaminase levels (AST/ALT <200 IU/L) | 31.1 |
| A patient with atypical pneumonia | 9.7 |
| A patient with culture-negative endocarditis | 42.7 |
| Q3) How many patients have you diagnosed with Q fever? | |
| None at all | 38.7 |
| 1–2 patients have been diagnosed with Q fever | 35.0 |
| 3–5 patients have been diagnosed with Q fever | 17.5 |
| 6–10 patients have been diagnosed with Q fever | 3.9 |
| >10 patients have been diagnosed with Q fever | 4.9 |
| Q4) If a patient who was suspected to have acute Q fever had a negative result in their first serological test, did you perform a follow-up serological test at the convalescent stage of the illness? | |
| Follow-up serological tests were rarely performed | 55.3 |
| A second serological test was performed if possible | 37.8 |
| A patient was strongly recommended to undergo a second serological test for Q fever | 6.9 |
| Q5) Why was a second serological test for Q fever not performed in suspected patients with Q fever who had initially negative serological result at the acute stage of illness? (select all applicable items) | |
| They were diagnosed with other diseases | 53.4 |
| Unable to follow-up the patient after clinical improvement | 53.4 |
| The need for a follow-up test to identify Q fever was considered low after symptom improvement | 52.4 |
| Q6) Do you repeat the clinical assessments and serological tests over 6 months to identify chronic Q fever after acute Q fever was diagnosed? | |
| No, clinical assessments and serological tests are not repeated | 40.8 |
| Yes, clinical assessments and serological tests are repeated once or twice | 36.9 |
| Yes, clinical assessments and serological tests are repeated for at least 6 months | 22.3 |
| Q7) Have you ever performed the serological test or the PCR test for Q fever in patients with culture-negative endocarditis, infected aneurysm, or vascular graft infection? | |
| No | 45.6 |
| Yes | 54.4 |

100,000 persons per year) and Chungnam (0.27 cases per 100,000 persons per year) showed a much higher incidence than the other areas. Although *C. burnetii* is widely distributed world-wide, except in New Zealand, human Q fever has been reported primarily in endemic areas such as Australia, France, and Germany [9,10]. Its annual incidence was 1.7–4.0 cases per

100,000 persons in Australia and 0.1–0.4 cases per 100,000 persons in France, Germany, and Bulgaria [11–13]. The Netherlands experienced a large-scale outbreak of Q fever with >4,000 cases between 2007 and 2010; prior to the outbreak, approximately 10–30 cases were reported annually [2]. The outbreak of Q fever in the Netherlands was strongly associated with dairy goat farms [14]. Although there is no clear explanation for the recent trend toward an increase in the number of cases in South Korea, this finding could be explained by two assumptions. First, if more diagnostic tests are performed because of increased disease awareness among the medical staff, more Q fever diagnoses can be expected. Second, the increased incidence in humans might also be associated with an increase in Q fever in animals. The number of Q fever serological tests that were requested by the KCDC increased by 5.1-fold between 2014 and 2017 [15]. Through our online survey, we can also verify that most physicians were fully aware of which patients needed to be tested for Q fever. However, increasing testing may not be the only reason for the increase in human Q fever cases in South Korea. Although the number of animals with Q fever in livestock farms was low, with <10 cases reported each year, this number has been increasing steadily since animal Q fever was first reported in 2013. In addition, the number of goat farms increased by 1.4-fold from 2012 to 2017, and there was a strong correlation between the number of goat farms and the incidence of human Q fever [15]. Although animal sources have not been identified in most Q fever cases, previous studies have suggested that epizootic Q fever in goat herds is associated with outbreaks in humans [16,17].

This study revealed that in South Korea, Q fever more commonly occurred in men in their 50s without any epidemiological risk factors. Although human Q fever is a zoonotic disease, most patients with Q fever did not have an occupational risk (24.0%) or a history of animal contact (23.0%), which increases the risk of exposure to *C. burnetii*. Given that the main route of transmission of Q fever is aerosol-type propagation through dust or wind, human Q fever cases can occur sporadically without direct occupational or animal exposure [3,10]. This result is similar to that reported by a French study, demonstrating that only 8% of acute Q fever cases were related to occupational risks and 35.4% cases were related to animal contact [18]. These epidemiological characteristics in patients with Q fever may have led to an underdiagnosis of acute Q fever.

The clinical manifestations in patients with acute Q fever are different in each country [18–21]. Hepatitis was the most common clinical manifestation of acute Q fever in South Korea. Although there is no obvious hypothesis to explain why most patients with acute Q fever presented with hepatitis in South Korea, a possible explanation can be the significant correlation observed in this study between the number of goat farms and the incidence in human Q fever in South Korea. It is known that hepatitis is more frequent clinical manifestation in sheep- and goat-breeding areas [22]. Although animal sources were not identified in most Q fever cases, the most common kind of animal contact was goats in this study, and there was an increase in the number of goat farms during the study period.

Among the patients with acute Q fever, seroconversion was identified in 27.3% patients through collection of sera at least twice during the acute and convalescent stage; 36.6% patients did not undergo a second serological test during the convalescent stage. Because seroconversion usually occurs 7–15 days after onset of symptoms in acute Q fever, it is necessary to follow-up patients with an antibody test using convalescent serum samples taken 3–6 weeks apart [23]. However, as shown in the results of our survey, more than half of the physicians did not follow-up patients using a serological test. PCR tests can be helpful in the diagnosis of acute Q fever in patients who developed symptoms within 2 weeks; the sensitivity of PCR tests has been reported to be 24%–98% [23]. This study also showed that PCR tests were useful for diagnosing acute Q fever in patients who had negative serological results at the acute stage. Although PCR testing for *C. burnetii* is not widely used to diagnose Q fever in South Korea, its use will enable clinicians to make a correct and early diagnosis of acute Q fever.

In South Korea, vascular infection was a more common clinical manifestation than endocarditis in patients with chronic Q fever. It was known that endocarditis is most common manifestation among patients with chronic Q fever. In some studies, *C. burnetii* infection was the most frequently identified etiology in patients with culture-negative endocarditis [24,25]. In this study, Q fever endocarditis was only confirmed in 6 cases. However, culture-negative endocarditis cases accounted for about a quarter of endocarditis cases in South Korea [26,27]. This suggests that Q fever endocarditis may be an underestimated disease in South Korea, and serological tests or PCR tests for *C. burnetii* should be more actively performed in cases of culture-negative endocarditis. This study revealed that patients with chronic Q fever were likely to undergo surgery (41.2%), such as valve replacement or vascular surgery, and that they had a high mortality (23.5%). These clinical burdens and features of chronic Q fever are well-known in France and the Netherlands [28,29]. In South Korea, arterial aneurysms, heart valve insufficiency, and prosthetic valves were also risk factors in patients with chronic Q fever. Although there were only a few identifiable cases of chronic Q fever, the study results suggest that it is important to screen patients with risk factors, follow-up patients using serological tests, and appropriately monitor clinical changes in patients with acute Q fever. A study reported that the risk of endocarditis reached 39% among patients with acute Q fever and heart valve disease [30]. Many experts recommend serological and clinical follow-ups for at least 6 months up to 2 years after acute Q fever, depending on the risk factors [3,23]. Some experts recommended that patients with acute Q fever and risk factors for chronic Q fever should receive prophylactic antibiotics for 1 year [23]. However, our results showed that a follow-up for 6 months to identify chronic Q fever was only performed by 24.0% of physicians and that only 41.0% of physicians performed >3 follow-up serological tests. Hence, chronic Q fever, despite its clinical severity, seems to be an underreported and neglected infectious disease.

The results of the survey on physician's awareness indicated that many physicians, including infectious disease specialists, did not know how to correctly diagnose and manage Q fever. This may be due to the nonspecific clinical features of Q fever and the limited diagnostic tools available. Q fever can lead to persistent localized infection in certain patients with risk factors, therefore, it is necessary for clinicians to increase their awareness regarding Q fever to enable correct diagnosis and management.

In conclusion, Q fever is becoming an endemic disease in the midwestern areas of South Korea, such as Chungbuk and Chungnam, and is mainly diagnosed in previously healthy men in their 50s living in rural areas without other epidemiological risk factors. *C. burnetii* infection can occur in susceptible persons living far from infected animals or contaminated environments because aerosol particles can travel with wind for miles. Therefore, clinicians should suspect Q fever in patients with acute hepatitis of unknown etiology in South Korea. In addition, given the upward trend of acute Q fever cases and the clinical severity of chronic Q fever, appropriate diagnostic and management practices should be followed after primary *C. burnetii* infection. The disease burden of chronic Q fever should be evaluated in patients with culture-negative endocarditis or infected aneurysms.

## Supporting information

**S1 Appendix. Survey of physician's awareness regarding the diagnosis and management of Q fever.**
(DOCX)

**S1 Fig. Distribution of Q fever cases by age groups and sex from 2011 to 2017.**
(TIF)

**S1 Table. Temporal distribution of Q fever cases from 2011 to 2017.**
(DOCX)

**S2 Table. Results of other questions included in the questionnaire other than the key questions.**
(DOCX)

## Acknowledgments

We thank the following individuals/collaborators for collecting the data: Kyung Mok Sohn (Chungnam National University School of Medicine, Daejeon), Hee-Sung Kim (Chungbuk National University Hospital, Chungbuk Nationa University College of Medicine, Cheongju), Sang-Rok Lee (Cheongju St. Mary's Hospital, Cheongju), Chang-Seop Lee (Chonbuk National University, Jeonju), Jae Hoon Lee (Wonkwang University College of Medicine, Iksan), Sook In Jung (Chonnam National University Medical School), Chisook Moon (Busan Paik Hospital, Inje University, Busan), Kye-Hyung Kim (Pusan National University School of Medicine, Busan), Kyung-Wook Hong (Gyeongsang National University College of Medicine, Jinju), Joon Young Song (Korea University Guro Hospital, Seoul), Min-Chul Kim (Chung-Ang University Hospital, Seoul) Joon Sup Yeom (Yonsei University College of Medicine, Seoul), Jong Hun Kim (Korea University Anam Hospital, Seoul), Eu Suk Kim (Seoul National University Bundang Hospital, Seongnam), Won Suk Choi (Korea University Ansan Hospital, Ansan), Eun Ju Jung (Hallym University College of Medicine, Chuncheon), Young Keun Kim (Yonsei University Wonju College of Medicine, Wonju)

## Author Contributions

**Conceptualization:** Hee Young Lee, Jung Yeon Heo.

**Data curation:** Yong Chan Kim, Hye Won Jeong, Dong-Min Kim, Kyungmin Huh, Sang-Ho Choi, Hee Young Lee, Yunjung Jung, Yeol Jung Seong, Eun Jin Kim, Young Hwa Choi.

**Formal analysis:** Yong Chan Kim.

**Funding acquisition:** Jung Yeon Heo.

**Investigation:** Yunjung Jung, Yeol Jung Seong, Eun Jin Kim, Young Hwa Choi.

**Methodology:** Hee Young Lee.

**Project administration:** Hye Won Jeong, Jung Yeon Heo.

**Writing – original draft:** Yong Chan Kim.

**Writing – review & editing:** Dong-Min Kim, Kyungmin Huh, Sang-Ho Choi, Eun Jin Kim, Young Hwa Choi, Jung Yeon Heo.

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
