## [Decision Letter · Decision Letter 0]

24 Dec 2020

Dear Dr. Heo,

Thank you very much for submitting your manuscript "Epidemiological investigation and physician awareness regarding the diagnosis and management of Q fever in South Korea, 2011 to 2017" for consideration at PLOS Neglected Tropical Diseases. As with all papers reviewed by the journal, your manuscript was reviewed by members of the editorial board and by several independent reviewers. In light of the reviews (below this email), we would like to invite the resubmission of a significantly-revised version that takes into account the reviewers' comments. 

We cannot make any decision about publication until we have seen the revised manuscript and your response to the reviewers' comments. Your revised manuscript is also likely to be sent to reviewers for further evaluation.

Sincerely,

Julie Arsenault, DVM, M.Sc., Ph.D.

Guest Editor

Hélène Carabin

Deputy Editor

Reviewer's Responses to Questions

**Key Review Criteria Required for Acceptance?**

**Methods**

-Are the objectives of the study clearly articulated with a clear testable hypothesis stated?

-Is the study design appropriate to address the stated objectives?

-Is the population clearly described and appropriate for the hypothesis being tested?

-Is the sample size sufficient to ensure adequate power to address the hypothesis being tested?

-Were correct statistical analysis used to support conclusions?

-Are there concerns about ethical or regulatory requirements being met?

Reviewer #1: (No Response)

Reviewer #2: -Are the objectives of the study clearly articulated with a clear testable hypothesis stated? Yes

-Is the study design appropriate to address the stated objectives? Yes

-Is the population clearly described and appropriate for the hypothesis being tested? Can be improved

-Is the sample size sufficient to ensure adequate power to address the hypothesis being tested? Yes

-Were correct statistical analysis used to support conclusions? Yes

-Are there concerns about ethical or regulatory requirements being met? Yes, see below.

Although the definition of confirmed and probable case of acute Q fever and chronic Q fever seem clear in this section, it is not completely congruent with the results section and it should be reviewed (see comments below).

The paragraph explaining the survey is redundant since all data are available in Table 4 and supplementary material, this paragraph could be summarized.

The authors say that the Institutional Review Board waived the requirement for written consent. Tis fact is hugely surprising, since Ethics Committees require it when it is necessary to consult the medical personal history of patients. To clarify this fact the authors could explain how the information was extracted avoiding the consultation of these documents.

Reviewer #3: The methods are well described and clear to the reader.

A few minor comments in this section - I am not aware of the definition of "probable case" as defined by the authors by a single IgG phase 2 >256. This titer might as well represent a past infection, and this should be stated to my opinion.

In addition, the survey to evaluate physicians' awareness of Q fever is in the appendix, and I think it is not necessary to write so many examples of questions in so many details. 

Otherwise the statistical analysis and ethics are well described.

**Results**

-Does the analysis presented match the analysis plan?

-Are the results clearly and completely presented?

-Are the figures (Tables, Images) of sufficient quality for clarity?

Reviewer #1: Concerning acute Q fever, the table reported is the one known in the literature Concerning "chronic Q fever". It must be clarified, as the authors are beginning to do. This term is becoming obsolete and that it must be replaced by resistant focalised infections, which means that the risk factors of its forms do not constitute risk factors but fixing factors. Valvulopathies or aneurysms are not risk factors for chronic Q fever but risk factors for chronic infection of aneurysms or valvulopathies. It would be interesting to reflect on the importance of Q fever in endocarditis with negative blood cultures in Korea and to see if there are systematic studies carried out for this purpose and this could be a suggestion in the discussion to make a systematic Coxiella burnetii serology or PCR on valves of endocarditis , negative blood culture which allows a true evaluation of the incidence. In the same way, the revelation by PET scanner of a fixation on a vascular prosthesis or an aneurysm is a formal indication to carry out an examination to detect Q fever. These elements could be usefully added. The whole discussion about the knowledge of Korean physicians is not of major interest to investigators in other countries. Concerning the different scenarios, Table 1 is not really of interest and could be deleted over, as could Table 4.

Reviewer #2: -Does the analysis presented match the analysis plan? Yes

-Are the results clearly and completely presented? Can be improved

-Are the figures (Tables, Images) of sufficient quality for clarity? Yes

It is important to confirm that the 200 patients are different, and no chronic Q fever case was included also as an acute Q fever case, since the study period could lead to this mistake.

Acute Q fever cases: the diagnosis criteria are very confusing regarding acute cases and in the present text form it seems that some patients had only a clinical diagnosis. The authors should re-write this paragraph and clarify which cases were diagnosed with a single serological test, seriated serological tests or only clinical criteria. These data should also be congruent with those shown in Table 2.

In the same way, the authors say that 67 patients achieved the diagnosis of probable Q fever. Did clinicians establish a clinical diagnosis and proper treatment in these cases? Did this subgroup of patients undergo to a correct clinical and serological follow up? In this sense it could be useful to use the criteria and recommendations wich appear in the reference 22 by Eldin et al.

Regarding clinical form of presentation it could be of a high interest to know if the presentation as hepatitis, isolated fever or pneumonia was more frequent depending on the living area of patients, as it happens in other countries like Spain, like it is shown in the reference 20 by Alende-Castro et al.

Can the authors add some information about the treatment and clinical evolution of patients diagnosed with acute Q fever?

Chronic Q fever cases: since this disease is infrequent it would be of interest if authors could show more information about the reported cases, for example, which type of vascular involvement showed the other cases apart from those 3 with aortic prosthesis related infection?

The authors say that they were 9 cases of vascular infection, but Table 3 shows 8.

Did any of the chronic Q fever cases had previous diagnosis of acute Q fever? If so, was it correctly treated? 

Survey: the paragraph regarding survey responses is redundant since most of the information is available in tables, it could be summarized.

Reviewer #3: The results are described very clearly in the text and in the tables and figures.

**Conclusions**

-Are the conclusions supported by the data presented?

-Are the limitations of analysis clearly described?

-Do the authors discuss how these data can be helpful to advance our understanding of the topic under study?

-Is public health relevance addressed?

Reviewer #1: This work reports the description of cases of Q fever in Korea which is a country for which this disease is not well known, some elements are contributory for the scientific community and others less obviously interesting. From this point of view, surveillance to assess the interest and knowledge of clinicians for Q fever can be removed and is of little interest.

Reviewer #2: -Are the conclusions supported by the data presented? Yes

-Are the limitations of analysis clearly described? No, the authors should add a specific paragraph

-Do the authors discuss how these data can be helpful to advance our understanding of the topic under study? Can be improved

-Is public health relevance addressed? Yes

Regarding discussion and conclusions, here are my comments:

The authors should make more emphasis in the relevance of their findings, in this sense, the clinical suspicion of Q fever in South Korea could be establish also in the absence of an epidemiological risk factor? Do the authors have any hypothesis to explain that most patients presented as hepatitis? Do the authors consider reevaluating the effect of their intervention (through the survey) in the clinical diary practice regarding Q fever?

The authors refer again to the fact that 27.3% of acute cases tested negative, PCR test were performed only in 15.8% and 36.6% did not undergo a second test. As it was recommended above, this paragraph should be reviewed and clarify all the subgroups of patients considering the diagnosis process.

What is the availability of Q fever PCR tests in South Korea? In many countries this test is not routinely available and could explain the low percentage of patients in which it was performed. It would also be of interest to describe the indications for PCR test if exist.

Regarding chronic Q fever cases, the authors discuss about the risk factors of patients (vascular prosthesis) but they did not explain in the results section if the diagnosed patients had any of these risk factors, both parts of the manuscript should be congruent.

Finally, the authors could discuss deeply the follow-up of patients in their series and the low relevance that physicians give to this process in light of the answers reflected in the survey.

Reviewer #3: The discussion and conclusions are derived from the results of the paper, are comprehensive and clear, and well written.

**Editorial and Data Presentation Modifications?**

Reviewer #1: (No Response)

Reviewer #2: In my opinion, the study is valuable but should be reviewed to facilitate its comprehension to readers.

Reviewer #3: The paper is well written, and the data is presented clearly.

The first paragraph of the introduction needs a reference for the statement that "...and has resulted in large-scaled outbreaks".

The second paragraph in the introduction section, line 5, should be rephrased to "Although the acute Q fever is USUALLY benign with a low mortality rate...", since acute Q fever is not always benign and these cases are well described in the literature.

**Summary and General Comments**

Reviewer #1: (No Response)

Reviewer #2: See comments above regarding specific sections. I want also to add this general comments:

The authors present an interesting work showing data about the increasing of Q fever cases in South Korea. The overall design of the study is correct, and the results could be of general interest, but some aspects must be clarified.

The authors could review some English expression along all the manuscript to improve reading comprehension.

The introduction section could be enriched if some references are added to support data, for example in the first paragraph when the authors refer to large outbreaks.

Reviewer #3: I think the paper is well written and designed.

The importance of the data is mostly local, for the South Korean community, but the paper has also global importance since the issues that are dealt in the paper, about awareness and need for follow up in high risk patients are extremely important everywhere, thus the paper should be accepted and published with the minor revisions I already mentioned.

PLOS authors have the option to publish the peer review history of their article (what does this mean?). If published, this will include your full peer review and any attached files.

Reviewer #1: No

Reviewer #2: No

Reviewer #3: No
---

## [Decision Letter · Decision Letter 1]

3 May 2021

Dear Dr. Heo,

Thank you very much for submitting your manuscript "Epidemiological investigation and physician awareness regarding the diagnosis and management of Q fever in South Korea, 2011 to 2017" for consideration at PLOS Neglected Tropical Diseases. As with all papers reviewed by the journal, your manuscript was reviewed by members of the editorial board and by several independent reviewers. The reviewers appreciated the attention to an important topic. Based on the reviews, we are likely to accept this manuscript for publication, providing that you modify the manuscript according to the review recommendations. 

Dear Authors,

Please add a sentence in the methods to clarify which statistical procedure was used to compare baseline characteristics of patients with acute versus chronic Q fever (in reference to Table 1).

Best regards,

Julie Arsenault

Sincerely,

Julie Arsenault, DVM, M.Sc., Ph.D.

Guest Editor

Hélène Carabin

Deputy Editor

Dear Authors,

Please add a sentence in the methods to clarify which statistical procedure was used to compare baseline characteristics of patients with acute versus chronic Q fever (in reference to Table 1).

Best regards,

Julie Arsenault

Reviewer's Responses to Questions

**Key Review Criteria Required for Acceptance?**

**Methods**

-Are the objectives of the study clearly articulated with a clear testable hypothesis stated?

-Is the study design appropriate to address the stated objectives?

-Is the population clearly described and appropriate for the hypothesis being tested?

-Is the sample size sufficient to ensure adequate power to address the hypothesis being tested?

-Were correct statistical analysis used to support conclusions?

-Are there concerns about ethical or regulatory requirements being met?

Reviewer #2: (No Response)

**Results**

-Does the analysis presented match the analysis plan?

-Are the results clearly and completely presented?

-Are the figures (Tables, Images) of sufficient quality for clarity?

Reviewer #2: (No Response)

**Conclusions**

-Are the conclusions supported by the data presented?

-Are the limitations of analysis clearly described?

-Do the authors discuss how these data can be helpful to advance our understanding of the topic under study?

-Is public health relevance addressed?

Reviewer #2: (No Response)

**Editorial and Data Presentation Modifications?**

Reviewer #2: (No Response)

**Summary and General Comments**

Reviewer #2: (No Response)

PLOS authors have the option to publish the peer review history of their article (what does this mean?). If published, this will include your full peer review and any attached files.

Reviewer #2: No

Figure Files:

Data Requirements:

Reproducibility:

References

---

## [Editor Report · Decision Letter 2]

11 May 2021

Dear Dr. Heo,

We are pleased to inform you that your manuscript 'Epidemiological investigation and physician awareness regarding the diagnosis and management of Q fever in South Korea, 2011 to 2017' has been provisionally accepted for publication in PLOS Neglected Tropical Diseases.

Best regards,

Julie Arsenault, DVM, M.Sc., Ph.D.

Guest Editor

Hélène Carabin

Deputy Editor

---

## [Editor Report · Acceptance letter]

27 May 2021

Dear Dr. Heo,

We are delighted to inform you that your manuscript, "Epidemiological investigation and physician awareness regarding the diagnosis and management of Q fever in South Korea, 2011 to 2017," has been formally accepted for publication in PLOS Neglected Tropical Diseases.

Best regards,

Shaden Kamhawi

co-Editor-in-Chief

Paul Brindley

co-Editor-in-Chief
